# MicroRNAs in Juvenile Idiopathic Arthritis: State of the Art and Future Perspectives

**DOI:** 10.3390/biology12070991

**Published:** 2023-07-12

**Authors:** Simone Pelassa, Federica Raggi, Chiara Rossi, Maria Carla Bosco

**Affiliations:** UOC Rheumatology and Autoinflammatory Diseases, Department of Pediatric Sciences, Istituto Giannina Gaslini, Istituto di Ricovero e Cura a Carattere Scientifico (IRCCS), 16147 Genova, Italy; simonepelassa@gaslini.org (S.P.); chiararossi@gaslini.org (C.R.); mariacarlabosco@gaslini.org (M.C.B.)

**Keywords:** microRNA, Juvenile Idiopathic Arthritis, Biomarkers

## Abstract

**Simple Summary:**

Juvenile Idiopathic Arthritis (JIA) is a chronic pediatric arthritis and an important cause of children’s disabilities. Despite major therapeutic advancements achieved in the last decades, the lack of diagnostic and predictive biomarkers still hinders JIA patients’ clinical management. MicroRNAs (miRNAs) are small nucleic acids whose role in many pathologies has been demonstrated. In particular, miRNAs circulating in patient body fluids represent potential new disease biomarkers. Although more extensively investigated in adult arthritis, the expression and roles of miRNAs have also been studied in JIA. Here, we review the most relevant studies analyzing the expression of miRNAs in patients affected by JIA and evaluate their potential as biomarkers, highlighting the limitations currently affecting miRNA investigation.

**Abstract:**

Juvenile Idiopathic Arthritis (JIA) represents the most common chronic pediatric arthritis in Western countries and a leading cause of disability in children. Despite recent clinical achievements, patient management is still hindered by a lack of diagnostic/prognostic biomarkers and targeted treatment protocols. MicroRNAs (miRNAs) are short non-coding RNAs playing a key role in gene regulation, and their involvement in many pathologies has been widely reported in the literature. In recent decades, miRNA’s contribution to the regulation of the immune system and the pathogenesis of autoimmune diseases has been demonstrated. Furthermore, miRNAs isolated from patients’ biological samples are currently under investigation for their potential as novel biomarkers. This review aims to provide an overview of the state of the art on miRNA investigation in JIA. The literature addressing the expression of miRNAs in different types of biological samples isolated from JIA patients was reviewed, focusing in particular on their potential application as diagnostic/prognostic biomarkers. The role of miRNAs in the regulation of immune responses in affected joints will also be discussed along with their potential utility as markers of patients’ responses to therapeutic approaches. This information will be of value to investigators in the field of pediatric rheumatology, encouraging further research to increase our knowledge of miRNAs’ potential for future clinical applications in JIA.

## 1. Introduction

### 1.1. Juvenile Idiopathic Arthritis: Clinical Features and Treatment

JIA is an umbrella term used to indicate a clinically heterogeneous group of pediatric chronic rheumatic disorders of unknown origin and etiology, with a prevalence of 20.5 cases per 100,000 population in Europe and onset before 16 years of age. The disease is characterized by distinct clinical manifestations, course, response to therapy, and probably genetic background [1,2,3,4,5,6,7,8]. Persistent synovial inflammation is an important feature of the joints of JIA patients [1,3,9,10,11], which may lead to alterations of their structure with consequent functional impairment and disability, and is associated with extra-articular inflammatory manifestations and severe complications (such as chronic uveitis or Macrophage Activation Syndrome) [3,12,13]. Nowadays, JIA patients are grouped into several different categories according to the 2001 International League of Associations for Rheumatology (ILAR) classification criteria: systemic arthritis (sJIA), oligoarthritis (oJIA), rheumatoid factor (RF)-positive polyarthritis (RF+ pJIA), RF-negative polyarthritis (RF− pJIA), enthesitis-related arthritis (ERA), psoriatic arthritis (PsA), and undifferentiated arthritis [14]. The specific characteristics of each disease subset are reported in Table 1.

In the last decade, limitations of the current classification have been pointed out, and the need for a revision of the ILAR criteria has emerged. In particular, a new homogeneous group of JIA patients showing several common features (such as early onset and antinuclear antibody (ANA) positivity), which includes JIA cases previously classified into different ILAR subtypes (oJIA, RF− pJIA, PsA), has been proposed and referred to as early-onset ANA-positive JIA [22,23,24]. The pharmacological treatment of JIA patients follows a step-up strategy dependent on the JIA category and disease activity. Non-steroidal anti-inflammatory drugs (NSAIDs) and intra-articular corticosteroid injections represent first-line therapies for all JIA subtypes. Non-responder patients require a second-line therapy and are treated with conventional disease-modifying anti-rheumatic drugs (DMARDs). Among these, methotrexate (MTX) is the most frequently used [3,12]. Corticosteroids can also be administered systemically for short periods of time in cases of symptoms resistant to other therapies or complications. The most severe cases of JIA are treated with new biological DMARDs, able to target the pro-inflammatory cytokines involved in the pathogenesis of the disease, such as anti-TNFα, anti-IL-1, anti-IL-6, and anti-CTLA4 antibodies, alone or in combination with MTX (Table 1) [3,15,20,21,25]. Despite the availability of various therapeutic options for the management of JIA, the majority of patients fail to achieve complete clinical remission but experience recurrent disease with relapse in treated joints, progressive spread to other joints, and/or severe complications by one or two years after onset [2,5,26,27,28,29]. Achieving sustained inactive disease and preventing disease extension, structural joint damage and complications are, thus, the main targets of new therapies [28]. However, early diagnosis and prediction of the JIA course/outcome, which are essential for tailoring therapeutic intervention, are hindered by the lack of validated biomarkers. In fact, although several studies have been carried out to identify new potential diagnostic markers, predictors of outcome, and indicators of disease activity [2,30,31,32,33,34,35], only a limited number of biomarkers have been included in routine clinical practice (Table 1). Specifically, anti-nuclear antibodies (ANA), rheumatoid factor (RF), HLA-B27 antigen, and anti-cyclic citrullinated peptide (CCP) antibodies are used for patient classification, while erythrocyte sedimentation rate (ESR) and C-reactive protein (CRP) are utilized for the definition of the Juvenile Arthritis Disease Activity Score (JADAS) [2,16,17,30], a composite scoring for the assessment of disease clinical activity [18,36]. Conversely, the Wallace criteria are used to identify patients with inactive disease and clinical remission on and off medication [26]. Hence, the identification of new low-invasive diagnostic/prognostic biomarkers measurable at an early stage of the disease remains highly critical to improve JIA patients’ clinical care.

### 1.2. microRNAs (miRNAs): Role in Disease Pathogenesis and as Potential Biomarkers

miRNAs are short non-coding single-stranded cellular RNAs (typically ≈ 20–22 nt) observed for the first time in *Caenorhabditis elegans* in 1993 [37,38] and later described in many other organisms, including mammals [39]. The expression of miRNAs is a tightly regulated process, and their biogenesis and mechanisms of action have been extensively characterized [19,40,41] and will not be further addressed in detail here. Briefly, miRNAs are generated through a maturation process driven by different molecules (the protein DGCR8 and the RNA endonucleases Drosha and Dicer), which leads to the formation of a double-stranded miRNA [42]. One of the two miRNA strands (the guide strand) is bound by the Argonaute (AGO) proteins loaded into the miRNA-induced silencing complex [39,40,43,44] and modulates cellular gene expression through the recognition of complementary regions within the mRNA target. Post-transcriptional inhibition of gene expression is the most reported and well-known mechanism of action of miRNAs, although under specific conditions they can also enhance gene expression [45,46,47,48]. Gene silencing is achieved through the recruitment of proteins that mediate mRNA deadenylation, decapping, and 5ʹ-to-3ʹ mRNA degradation [40,43]. miRNAs can be released by the cells in human body fluids (e.g., serum, plasma, urine, synovial fluid, saliva, tears, breast milk, pleural, peritoneal, cerebrospinal, and seminal fluids) [49,50,51,52] either as “free-circulating” molecules (bound to AGO2 or high-density lipoproteins) or encapsulated within extracellular vesicles (EVs), such as exosomes or microvesicles [40,53,54,55,56], a group of lipid bilayer membrane-delimited nanometer-sized vesicles acting as important mediators of intercellular communication [57,58,59]. Differences in the profile of “free-circulating” and intravesicular miRNAs (EV-miRs) have been reported, probably dependent on the higher resistance of EV-miRs to environmental conditions and protection from endogenous enzymatic degradation compared to “free” molecules. These characteristics result in increased EV-miR enrichment and stability [60] and, thus, higher detectability of low abundance molecules [55,61] and better reflection of the disease state [62,63,64].

In the last decades, miRNAs have become the focus of intensive investigations ranging from basic biology to clinical applications given their role in modifying gene expression and in the epigenetic regulation of multiple physiologic processes fundamental to human health, including inflammatory responses and immune cell differentiation, maturation, and functions [65,66]. In addition, since their initial discovery, numerous studies have reported miRNA involvement as a key mediator in the pathogenesis of many human diseases [67,68,69]. Altered miRNA expression profiles have, in fact, been closely linked to the development and progression of several types of cancers: hematologic, cardiovascular, respiratory, neurologic conditions, inflammatory, and autoimmune disorders [56,70,71,72,73,74,75,76]. miRNAs in patient biological samples are also currently under study for their potential as novel biomarkers [77,78,79,80,81,82]. In particular, recent reports have indicated the diagnostic/prognostic value of dysregulated miRNAs in patients with different types of rheumatic diseases [83,84,85,86]. However, miRNA investigation and application in clinical practice are hindered by the lack of consensus regarding the technique to be used for their isolation [87,88,89] and quantification [90,90,91,92,93]. Many kits for miRNA isolation from biofluids are based on the use of acid guanidinium thiocyanate-phenol-chloroform, which guarantees high efficiency but results in the loss of miRNAs with low GC content [89]. Several alternatives, such as the use of column-based clean-up after phenol or phenol-free techniques, are currently being used by different researchers, although with unclear performances [93,94,95]. Similarly, miRNA level quantification can be carried out using different techniques. Northern blotting was considered the “gold standard” in the past, whereas nowadays microarray, qRT-PCR, and NGS are commonly used by different groups to measure miRNA expression with advantages and disadvantages of each method [90,91,92]. Divergences in data normalization and the lack of standardization of the statistical approaches applied, such as, for instance, the choice of different cutoff values and of robust controls, represent other important drawbacks in miRNA analysis [90,96,97,98,99].

## 2. miRNAs in JIA: General Considerations

To date, the role of miRNAs in JIA has been investigated only by a few research groups, and a small number of papers are currently available in the literature on this issue (Table 2).

Comparison among published data is quite complicated because JIA is a rare disease; therefore, the number of patients enrolled in the studies is usually small, and results often have poor statistical significance. In addition, JIA patients analyzed in the different reports were characterized by great variability in terms of both clinical features and therapeutic regimens. Finally, different types of biologic samples (serum, plasma, synovial fluid) or miRNAs (cell-associated vs. released in body fluids as free-circulating or EV-associated molecules) were evaluated, and diverse experimental approaches for miRNA isolation and analysis were used in distinct studies (Table 2). 

In this review, we provide an update of the state of the art of the research on miRNA expression in different biologic samples from JIA patients and discuss recent advances in the understanding of their role as potential diagnostic/prognostic biomarkers. We performed MEDLINE, Scopus, and Google Scholar searches for “miRNAs AND Juvenile Idiopathic Arthritis”, “biomarkers in JIA”, and “miRNA AND biomarkers” (title/abstract). In particular, we focused on the clinical characteristics of JIA patients (subtypes, disease activity, and therapies), the type of biological samples used in the different studies, and the biological characteristics of the miRNAs analyzed (free or EV-derived). The comparison between data obtained in JIA vs. adult arthritides, as well as miRNA role in JIA pathogenesis, which have been previously reviewed by others [19,112], was not further addressed here. Finally, we summarized data in the literature highlighting the relevance of miRNAs as therapeutic biomarkers in patients affected by chronic inflammatory rheumatic conditions other than JIA to emphasize the need for future research in JIA aimed at assessing miRNA potential value as predictors of responses to therapies for clinical applications in this disease.

## 3. Cell-Associated miRNAs in JIA

### 3.1. Differentially Expressed miRNAs

Lashine and colleagues were the first group to evaluate miRNA expression in samples from children affected by JIA in 2015 [100]. Although the authors focused primarily on the assessment of mir-155 expression in mononuclear cells isolated from the peripheral blood (PBMCs) of children affected by systemic lupus erythematosus (SLE), 10 JIA patients were also analyzed in parallel. TaqMan RT-qPCR performed on total RNA showed that miR-155 expression was up-regulated in PBMCs from JIA patients as compared to 15 age- and sex-matched healthy controls (HCs). However, a limitation of this study is the absence of detailed clinical and laboratory information on enrolled JIA patients, such as disease category, number of affected joints, disease activity, and therapy at the time of enrollment, which hampers data interpretation.

In the same year, Kamiya and colleagues [101] analyzed PB leukocytes (PBLs) isolated from 6 JIA patients (3 sJIA and 3 pJIA) and 3 HCs. RT-PCR was carried out on total RNA derived from the cells to quantify the levels of expression of 5 selected miRNAs (miR-16, miR-132, miR-146a, miR-155, and miR-223). No differences in the expression of miRNAs (including miR-155) were reported in patients compared to HCs. Although patient classification according to the ILAR guidelines is reported in this paper, no further information concerning the treatment received by the patients or disease activity and stage is available.

Microarray analysis was carried out by Schulert and his group [102] to compare miRNA expression profiles in monocytes (Mn) from PB of 7 patients with active new-onset (NOS), 9 with established (clinically active disease despite treatment), 11 with clinically inactive (CID) sJIA, and 13 HCs. NOS patients were enrolled immediately after diagnosis and were treatment-naïve. Patients with established diseases were characterized by active arthritis with rash, fever, adenopathy, hepatosplenomegaly, and elevated levels of ESR or CRP, while CID patients were identified based on the Wallace Criteria [26,102]. A total of 110 miRNAs, including miR-155, miR-146a, miR-23a, and miR-27a, were reported to be more expressed in Mn from active (both with NOS and established disease) compared to CID patients and HCs. Interestingly, the expression of four miRNAs (miR-16, miR-24, miR-186, and miR-342-3p) in Mn positively correlated with serum levels of ferritin. miR-99a and miR-100 were down-regulated, while miR-133a was up-regulated, in Mn from NOS patients compared to those with established disease. In addition, the authors identified 37 miRNAs significantly modulated (31 up- and 6 down-regulated) in Mn of CID patients compared to HCs. Of the 31 up-regulated miRNAs, 21 were further increased in patients with both NOS and established active sJIA as compared to CID patients, pointing to their role in Mn activation during disease progression, whereas the other 10 miRNAs (including miR-223) were up-regulated in CID compared to HCs. The authors focused in particular on miR-125a-5p, known for its role in driving M2-like macrophage polarization [113], showing high expression in Mn from patients with active (both established and NOS) sJIA compared to CID patients and active pJIA patients, who were enrolled in parallel, suggesting expression specificity for active sJIA. Although no correlation between mir-125a-5p levels in Mn and active joint counts or inflammatory markers was observed, mir-125a-5p was found to be correlated with serum ferritin levels and platelet and PB white cell counts. miR-125a-5p was, thus, proposed as a potential biomarker for systemic inflammation rather than the degree of arthritis. It is important to point out that patients with active or inactive diseases were subjected to treatment with different drugs (MTX, biologic DMARDs, and/or corticosteroids), which may have affected miRNA expression, hampering comparisons with results from other works.

The differential expression of miR-146a in Mn from sJIA patients was also observed by Li et al. [103]. In this study, CD14+ Mn was isolated from the PB of 32 patients affected by sJIA (8 with NOS, 11 with established, and 13 with CID disease) and age- and sex-matched HCs, and a TaqMan microRNA assay was carried out. Mir-146a levels were found to be higher in active sJIA patients (both with NOS and established disease) compared with HCs.

The involvement of miR-125 family members in JIA was also reported by Fan et al. [104], who investigated miR-125b expression in PBMCs and CD4+ T lymphocytes from 16 RF+ treatment-naïve pJIA patients with early active disease compared to 22 age- and sex-matched HCs. Unfortunately, little clinical information on enrolled patients was provided in this study. TaqMan qPCR results showed significantly decreased levels of miR-125b in both patient-derived PBMCs and CD4+ cells with respect to HCs. Given the role of miR-125b in T-cell differentiation, the authors investigated the correlation between its expression in PBMCs and the Th17/Treg ratio, observing an increased number of Th17 and decreased frequency of Treg in patients compared to HCs. A negative correlation between miR-125b expression and ESR and CRP values was also reported.

A comparison of the miRNA expression patterns of PBMCs and paired mononuclear cells isolated from the synovial fluid (SFMCs) of 9 oJIA patients vs. PBMCs from 8 HCs was carried out by Rajendiran and colleagues [105]. Enrolled children received NSAID and 2 of them also received MTX before recruitment. Global miRNA analysis by gene chip array identified a group of 234 miRNAs differentially expressed in SFMCs with respect to paired PBMCs and PBMCs from HCs. Among them, miR-23a-3p, miR-23a-5p, miR-27a-5p, miR-146a-5p, and miR-155-5p, which were involved in the oxidative stress pathways, were significantly up-regulated in patients with SFMCs compared to PBMCs from HCs. Moreover, 7 miRNAs (miR-15a, miR-20a, miR-18a, miR-17-3p, miR-3652, miR-19b, and miR-19a) were up-regulated in patients with PBMCs compared to paired SFMCs and PBMCs from HCs. Additional analysis by RT-qPCR on independent samples (PBMCs and SFMCs from 6 oJIA patients and PBMCs from 8 HCs) confirmed significantly increased levels of miR-155-5p, miR-23a-3p, miR-27a-5p, and miR-146a-5p in patient SFMCs than in PBMCs from HCs. miR-155-5p and miR-146a-5p levels were also significantly higher in patient SFMCs with respect to paired PBMCs, while miR-23a-5p and miR-27a-5p in patients PBMCs compared to PBMCs from HCs [105].

Recently, McAlpine’s group carried out RT-PCR to measure the expression of miRNAs in PBLs from 40 patients diagnosed with active oJIA (37 persistent and 3 extended) [106]. The patients enrolled in this study were mostly ANA+ (82%) and treatment naïve or treated with NSAIDs only (45 and 55%, respectively). The comparison between PBLs from oJIA patients and 10 adult HCs did not reveal any differentially expressed miRNAs. On the contrary, miR-21-5p, miR-27a-3p, miR-146b-5p, and miR-155-5p were up-regulated in leukocytes isolated from SF compared to paired PBL, while miR-409-3p and miR-451a were down-regulated. No significant differences were observed between PBL from HCs and leukocytes from the blood and SF of oJIA patients.

### 3.2. Data Comparison among Different Studies

The comparison among studies from different groups shows both similar and divergent results (Table 2), which may probably be ascribed to the distinct clinical features of enrolled patients, the different techniques used for miRNA isolation and quantification, and the heterogeneity of the cell populations analyzed (PBMCs, PBL, Mn, T lymphocytes, SFMC, SF leukocytes). In particular, the PBMC and SFMC populations are composed only of mononucleate cells, while PBLs and SF- leukocytes also include neutrophils.

miR-155 modulation has been studied by several authors [100,101,102,105,106]. On the one hand, Lashine [100] and Schulert [102] observed miR-155 up-regulation in PBMCs and Mn from the PB of sJIA patients compared to HC cells, respectively. On the other hand, Kamiya [101] and Rajendiran [105] did not report any difference in terms of miR-155 expression in PBL and PBMCs from sJIA/pJIA and oJIA patients, respectively, compared to HCs. Furthermore, both Rajendiran [105] and McAlpine [106] observed higher expression of miR-155-5p in SFMC [105] and SF leukocytes [106] from oJIA patients compared to paired PBMCs and PBLs, respectively.

miR-146a was up-regulated in Mn of sJIA patients with active disease compared to inactive and healthy controls by both Schulert [102] and Li [103] but also in SFMCs from oJIA patients compared to paired and HC PBMCs by Rajendiran [105]. These data are in contrast with data by Kamiya’s study in which no differences in miR-146a expression were reported [101].

miR-23a and miR-27a were up-regulated in Mn from sJIA with active disease with respect to inactive patients and HCs by Schulert [105] but also in oJIA patient SFMCs and PBMCs compared to HC PBMCs by Rajendiran and colleagues [105]. In contrast, McAlpine’s group [106] did not report any difference in miR-27a-3p expression between PBLs from oJIA patients and HCs; however, it is important to point out that HCs enrolled in McAlpine’s study were adult individuals.

## 4. miRNAs Released in the Serum and PL of JIA Patients

### 4.1. Differentially Expressed miRNAs

The first study investigating miRNA expression in the serum of JIA patients was carried out by Kamiya et al. in 2015 [101]. Serum was collected from a total of 24 JIA patients (8 with sJIA and 16 with pJIA) and 8 healthy donors, among whom 14 patients (8 with sJIA and 6 with pJIA) were analyzed in both the active and inactive phases of the disease, while the other 10 pJIA patients were analyzed exclusively during the active (1 patient) or inactive (9 patients) phases. Five selected miRNAs (miR-16, miR-132, miR-146a, miR-155, and miR-223) were analyzed using RT-qPCR. In contrast to what was reported in PBLs, the authors observed differential expression of these miRNAs in the serum. First, miRNA expression was compared among active and inactive sJIA and pJIA patients and HCs. miR-223 expression was significantly higher in the serum of active with respect to inactive pJIA and sJIA patients. In addition, active sJIA patients showed higher levels of miR-223 than inactive pOJIA patients and HCs. miR-132 expression was found to be significantly higher in both active and inactive sJIA patients in comparison to inactive, but not active, pJIA patients. No significant differences were reported in comparison to HCs. miR-155 was found to be significantly more expressed but only in active sJIA when compared to inactive pJIA patients. Lower, although not significant, miR-155 levels were observed in inactive than in active sJIA patients. Finally, no significant differences were observed in miR-146a levels. The authors then investigated the changes in miRNA expression during the different phases of the disease. Specifically, miRNA levels were measured in serum samples collected from 8 sJIA and 6 pJIA patients during both the active and inactive phases of the disease. miR-223 expression was significantly higher during the active phase in 6 out of 8 sJIA patients, lower in 1, while 1 did not show any difference. No significant differences were observed in pJIA patients. In addition, the potential of miRNAs as predictive biomarkers in JIA was investigated, observing a correlation between the serum levels of a few of them and some patient laboratory features. In particular, regression analysis showed a correlation between miR-223 expression and ESR levels and between miR-16 expression and MMP3 concentration in the serum of both sJIA and pJIA patients. A correlation was also found between serum MMP3 levels and both miR-146 and miR-223 expression in pJIA patients. Thus, miR-223 levels in serum have been suggested to represent a potential biomarker useful for the evaluation of disease activity in JIA. Although the authors reported some clinical characteristics of the enrolled patients, other important information that could have been useful for study correlation, such as positivity for ANA and the therapy followed by the patients during the study, are missing.

In the same year, Ma and colleagues [107] identified by microarray analysis 56 miRNAs differentially expressed in the PL of 5 oJIA and 5 pJIA patients at onset compared to 3 age- and sex-matched HCs. Among them, 5 miRNAs (miR-16, miR-146-a, miR-155, miR-223, and miR-132) were selected according to their putative role in immune pathways and further analyzed by RT-qPCR in PB samples from an independent cohort of 139 children (43 oJIA, 37 pJIA, 29 Juvenile Ankylosing Spondylitis (JAS), and 30 HCs). Enrolled JIA patients had predominantly active disease (79.1% oJIA and 94.6% pJIA), and about 50% were ANA+ (48.8% oJIA and 40.5% pJIA). Half of patients had less than 1 year of disease course (48.8 and 40.5% among oJIA and pJIA patients, respectively), while the other patients were diagnosed two years before enrollment. Second-line drug therapy was administered to 23.3 and 29.4% of oJIA and pJIA patients, respectively. Results showed that miR-146a, miR-16, and miR-223 were up-regulated, while miR-132 was down-regulated in PL from oJIA and pJIA patients compared to JAS and HC subjects. miR-155 was significantly down-regulated in PL from oJIA patients compared to HCs but not JAS patients, while down-regulation in pJIA patients was significant with respect to both JAS and HCs patients. No difference in miRNA expression levels was observed by comparing PL samples from oJIA and pJIA patients, except for miR-16, whose levels were significantly higher in the latter. ROC curves suggested that the levels of expression of four analyzed miRNAs (miR-16, miR-146a, miR-223, and miR-132) may have potential value for the diagnosis of JIA. In particular, miR-16 and miR-146a were effective at discriminating JIA patients from HCs but not oJIA from pJIA patients. Interestingly, miR-16 and miR-146 PL levels seemed to have a trend of correlation with IL-6 and TNF-α cytokine concentrations in patients with PL. Finally, the relationship between miRNAs and clinical parameters was examined. The authors reported that miR-16 levels in PL samples correlated with a few disease scores, such as the Juvenile Arthritis magnetic resonance imaging scoring (JAMRIS) for the hip and JADAS-27. Moreover, the levels of miR-146a in PL samples seemed to positively correlate with the JAMRIS for the hip, the limited joint count (LTC), and the 10-joints/27-joints/71-joints JADAS (JADAS-10/27/71), while PL miR-223 levels showed a decreasing trend with wrist or knee involvement. These data suggest the possible role of the PL levels of these miRNAs as biomarkers of JIA progression.

Later on, Demir and colleagues [108] quantified by RT-qPCR the expression levels of 4 selected miRNAs, given their reported role in the pathogenesis of autoimmune diseases. The levels of miR-16, miR-155, miR-204, and miR-451 were measured in the PL of 31 JIA patients (17 oJIA, 9 pJIA, and 5 ERA) and 31 HCs. Samples were collected from patients during both their active and inactive phases of the disease. While patients had not received any therapy during the active phase, they were treated with MTX for six months during the inactive period, and 6 of them received anti-TNF-α agent in combination with MTX. The authors reported higher miR-16 levels in patients during both their active and inactive periods compared to HCs, although no statistical significance was observed. No significant differences in miR-16 levels were reported among the oJIA, pJIA, and ERA patients. Moreover, no differences in terms of miR-16 expression between active JIA patients with normal and high acute phase reactant levels were detectable. Given the elevated miR-16 levels in the PL of patients affected by JIA, the authors suggested a role for this molecule in the pathogenesis of the disease and proposed miR-16 as a potential diagnostic biomarker. Although miR-155 was higher in patients during both the active and inactive phases of the disease compared to HCs, significant differences were observed only between PL in the active and inactive JIA patients. miR-155 levels in JIA patients treated with only MTX during the inactive period were significantly elevated in comparison to both JIA patients in the active phase and HCs. Higher miR-155 levels were reported in the PL of pJIA patients with respect to HCs, although the results had no statistical significance. miR-204 was found to be higher in the PL of active compared to inactive JIA patients and lower with respect to HCs, while levels in the PL of inactive JIA patients were lower than in HCs. However, no statistical significance was observed. In contrast, significantly lower levels were observed by comparing miR-204 PL levels in inactive JIA treated with MTX and HC individuals. In addition, no significant differences were reported between pOJIA, oJIA, and ERA. Finally, levels of miR-451 were increased in ERA and oJIA as compared to pJIA but without statistical significance.

Sun et al. [109] identified 48 miRNAs differentially regulated in the PL of 5 new-onset sJIA patients compared to 5 HCs by microarray analysis. Two of them in particular, miR-26a and miR-145, were further investigated by comparing their expression levels in PL from 20 sJIA vs. 30 JAS, 25 SLE, 40 KD, 40 Henoch-Schönlein purpura (HSP), and 40 HCs. All patients were enrolled at disease onset, had active disease, and were treatment-naive. Both miRNAs were found to be significantly up-regulated in sJIA compared to the other groups of the patients and HCs. To determine their specificity for the sJIA subtype, miR-26a and miR-145 expression levels were also measured in PL samples from 40 oJIA and 25 pJIA patients. The results demonstrated that only miR-26a was differentially expressed in sJIA compared to the other subtypes, while no difference was observed in oJIA- and pJIA-compared to HC-derived samples. ROC analysis was then performed to investigate whether miR-26a had diagnostic potential in sJIA. AUC showed that PL levels of miR-26a could discriminate sJIA patients from HCs, from children affected by different rheumatic diseases, such as KD and HSP, and from individuals affected by different JIA subtypes, such as oJIA and pJIA, thus confirming its diagnostic value in sJIA. Positive correlations between miR-26a levels and IL-6 concentrations in the PL of patients with sJIA were also observed, although no correlation between miR-26a and CRP and ESR was reported. 

Mc Alpine’s group compared miRNA expression levels in pools of cDNA from PL of 5 oJIA and 5 HCs subjects by RT-ddPCR using a 84-targeted microRNA PCR array, identifying 11 differentially expressed miRNAs (miR-15a-5p, miR-15b-5p, miR-16-5p, miR-24-3p, miR-126-3p, miR-192-5p, miR-195-5p, miR-223-3p, miR-324-5p, miR-451a, miR-602) [106]. The miRNAs showing higher differential expression and other miRNAs selected from the literature were further quantified in matched SF and PL samples from 20 patients. Results showed that miR-15a-5p and miR-409-3p were significantly higher in PL from pJIA patients compared to PL from HCs [108].

Finally, no differentially expressed miRNAs were identified by Nziza et al. [110] by high throughput next-generation sequencing (NGS) in the serum from 18 oJIA patients with respect to serum from 16 septic arthritis (SA) patients used as a control.

### 4.2. Data Comparison among Different Studies

Similar to what was observed for miRNAs isolated from cell populations, comparison among findings from different studies reveals both comparable and different miRNA expression patterns (Table 2).

Higher levels of miR-223 were observed by Ma [107] and Kamiya [101] in PL from pJIA and oJIA patients and serum from active pJIA and sJIA patients, respectively, compared to the specific HCs, although differences in the serum were not statistically significant. miR-132 was found to be significantly lower in PL of pJIA compared to HCs by Ma et al. [107], while no differences were observed by Kamiya et al. [101] among the sera from sJIA and pJIA patients and HCs. In contrast, miR-146a expression was found to be significantly elevated in the PL of pJIA patients [107] but not in the serum of sJIA and pJIA patients [101], with respect to HCs. These differences could be attributable to the different biological samples analyzed (PL vs. serum) [114] and the clinical features of JIA patients.

Interestingly, Ma [107] reported statistically significantly lower levels of miR-155 in PL samples from pJIA and oJIA compared to HCs, whereas Kamiya [101] did not observe any significant difference in miR-155 expression between serum from pJIA patients and HCs. These observations differ from those by Demir et al. [108] who reported that miR-155 expression was higher (although not statistically significant) in PL of oJIA, pJIA, and ERA patients compared to PL from HCs.

As shown by Ma et al. [107], miR-16 was found to be significantly up-regulated in the PL of oJIA and pJIA patients compared to HCs, while no statistically significant differences were reported by Demir et al. [108] in oJIA, pJIA, and ERA patients during both their active and inactive disease compared to HCs. Mc Alpine’s group results [106] are consistent with results from Ma and colleagues [107], confirming that miR-16 PL expression levels may be able to distinguish oJIA patients from HCs.

Finally, miR-451 levels were found to be increased in ERA and oJIA as compared to pJIA patients by Demir et al., although without statistical significance [108], while McAlpine et al. observed miR-451 up-regulation in PL of oJIA patients compared to HCs.

## 5. miRNAs Released in SF from JIA Joints

### 5.1. Differentially Expressed miRNAs

miRNA expression in SF from JIA patients was first reported in 2020 by Nziza et al. [110], who used RT-PCR and NGS to measure miRNA levels in SF collected from 18 oJIA and 16 SA patients. Enrolled individuals had not received previous treatment with steroids, DMARDs, or biological therapy. The authors first used an exploration cohort of centrifuged samples (5 sJIA and 3 SA patients) to identify miRNAs differentially expressed in JIA vs. SA patients. PCA results showed the differential expression of 141 miRNAs, narrowed down to 21 using stricter analysis criteria (miR-146a-5p, miR-150-5p 5, miR-155-5p, miR-2909, miR-339-3p, miR-342-5p, miR-4419a, miR-4419b, miR-4646-5p, miR-4667-5p, miR-4800-5p, miR-6716-5p, miR-6734-3p, miR-6764-5p, miR-6782-5p, miR-6794-5p, miR-6841-3p, miR-7150, miR-8063), which could discriminate between JIA and SA patients. Among them, 16 were up-regulated and 5 down-regulated. These results were then analyzed by HTG EdgeSeq on a validation cohort of samples, not centrifuged before storage, derived from 4 JIA and 4 SA patients. Of the 21 identified miRNAs, 19 were confirmed as differentially expressed between oJIA and SA SF, while miR-3687 and miR-4417 differential expression was not confirmed. Further validation of the 19 miRNAs was then carried out by RT-qPCR on a new independent cohort of samples centrifuged before storage to rule out possible differences due to sample processing (9 oJIA and 9 SA patients). Four miRNAs were found to be highly up-regulated (miR-155, miR-150-5p, miR-146a-5p, miR-342-5p) and 1 down-regulated (miR-6764-5p) in SF from oJIA compared to SF from SA patients. ROC curves and AUC values suggested the critical value of these miRNAs, particularly the combination of miR-6764-5p, miR-155, and miR-146a-5p, as putative diagnostic biomarkers able to distinguish JIA from SA patients.

The potential role of miRNAs released in SF of JIA patients as diagnostic biomarkers was then investigated by Mc Alpine et al. [106]. The authors compared miRNA expression in unmatched SF and PL of oJIA patients and 20 PL of HC subjects by ddPCR (for patient information, see Chapter 4). First, miRNAs were evaluated in pools composed of equal volumes of cDNA from SF or PL from 5 patients. Fourteen miRNAs (miR-15a-5p, miR-16-5p, miR-21-5p, miR-24-3p, miR-125b, miR-146a, miR-192-5p, miR-195-5p, miR-382-5p, miR-451a, miR-484, miR-513a-5p, miR-602, miR-654-5p) were differentially expressed between pooled SF and PL samples from oJIA patients. miRNAs showing higher differential expression, along with other miRNAs selected from the literature, were then quantified in 20 matched SF and PL samples. MiR-21-5p, miR-27a3p, miR-146b-5p, miR-155-5p, and miR-423-5p were significantly higher in SF samples compared to PL samples, whereas miR-192-5p and miR-451a were significantly decreased. In SF, miR-155-5p expression was correlated with disease duration, while miR-423-5p expression was inversely correlated with ANA positivity.

### 5.2. Data Comparison between Studies

SF reflects the biological milieu of the joint and may thus offer a direct measure of its pathologic state [19,84,115]. However, to date, miRNA expression in the SF of JIA patients and potential as biomarkers has been scarcely investigated.

As reported in Table 2, both groups addressing miRNA expression in SF analyzed samples from oJIA patients. However, the studies differed with regard to the samples used for comparison: Nziza and colleagues [110] utilized SF from SA patients, whereas McAlpine and colleagues [106] used paired patient PL. Interestingly, miR-155-5p, miR-146a-5p, and miR-146b-5p up-regulation in SF from oJIA patients was observed by both authors [106,110]. Furthermore, miR-27a overexpression in SF of oJIA patients found by McAlpine [106] was in line with data by Rajendiran [105], who showed its up-regulation in SFMCs with respect to PBMCs from oJIA patients, indicating the possible role of this miRNA in joint inflammation and pointing to SFMCs as its putative source in SF.

## 6. JIA-Derived EV-miRs

A great research effort in recent years has focused on the analysis of EV cargo to identify new predictors of disease development/evolution and response to therapy in several clinical settings [55,62,97,116,117,118,119,120,121], including in patients affected by adult arthritides [56,86,122]. However, EV-miR expression and potential as biomarkers in JIA have begun to be analyzed only in the last year. (Table 2). 

McAlpine and colleagues [106] investigated miRNA expression in EVs isolated by ultracentrifugation from the SF and PL of 20 new onset oJIA patients and the PL of 20 HCs. Interestingly, as assessed by cytofluorimetric analysis, EVs from the PL expressed the platelet marker CD142, while positivity for CD3 and CD19 (T and B lymphocyte markers, respectively) was significantly higher in EVs from SF than paired PL or HCs PL, suggesting a different cellular origin of EVs in the two distinct biological fluids. A statistically significantly higher number of EVs was present in PL but not SF from patients compared to PL from HCs. The levels of EV-miRs were quantified by RT-ddPCR, and the fraction of miRNA circulating in EVs from PL and SF samples was calculated. miR-155-5p was the miRNA found predominantly encapsulated in EVs (40.5–43.6% of the total amount), while miR-146b-5p, miR-192-5p, miR-423-5p, and miR-451a were present preferentially as free molecules.

The EV-miRNA expression profile in oJIA patients was investigated more extensively by Raggi and collaborators [111], who isolated EVs from PL and SF of 13 oJIA patients at disease onset and PL of 8 age- and sex-matched children undergoing minor orthopedic procedures enrolled as controls. Patients had not received any treatment before enrollment. miRNAs were isolated by membrane-affinity columns, and their profiles were analyzed by TaqMan Array Card Technology and compared among the different groups of samples. Differential expression analysis identified 24 and 79 EV-miRs that were significantly up- and down-regulated, respectively, in paired SF vs. PL samples of oJIA patients. In particular, a subset of 15 significantly deregulated EV-miRs, 7 up- (let-7c-5p, miR-21-5p, miR-34a-5p, miR-125b-5p, miR-155-5p, miR-193b-3p, miR-218-5p), and 8 down-regulated (let-7a-5p, let-7e-5p, let-7g-5p, miR-16-5p, miR-17-5p, miR-20a-5p, mir-26b-5p, and miR-106b-5p) was defined targeting multiple genes involved in biological processes related to inflammation responses, cartilage/bone degradation, and cell damage, suggesting their potential as biomarkers of OJIA development or candidate therapeutic targets. In addition, by comparing EV-miR expression profiles between samples from oJIA patients and HCs, the authors identified 110 (106 up- and 4 down-regulated) and 54 (25 up- and 29 down-regulated) EV-miR differentially expressed in patients PL and SF compared to HCs PL, respectively. A subset of 16 EV-miRs (let-7c-5p, mir132-3p, miR-146a-5p, mir-146b-5p, mir-186-5p, mir-210-3p, mir-21-5p, mir-222-3p, mir-24-3p, mir-29a-3p, mir-354-5p, mir-362-5p, mir-532-3p, mir-574-3p, mir-590-5p, mir-99a-5p) was consensually up-regulated in PL and SF samples from OJIA patients compared to HCs and able to significantly discriminate new-onset OJIA patients from HCs. ROC analysis demonstrated the high diagnostic potential of these miRNAs, which were suggested to represent a disease-associated molecular signature measurable at both local and systemic levels.

## 7. Concluding Remarks

In this review, we summarized some general trends emerging from published data that highlight miRNA’s potential as a candidate biomarker for earlier disease detection and outcome prediction, suggesting their putative clinical application to improve patient management. As pointed out above, the number of studies reported in the literature is limited, and differences exist in the technical approaches used for miRNA isolation and quantification and in the subtypes of patients analyzed (Table 2). However, a small group of differentially expressed miRNAs frequently recurring among the studies can be identified, such as miR-16, miR-125, miR-132, miR-146a, miR-155, and miR-223 refer to (Graphical Abstract), which highlights their potential relevance in JIA. The identification of extracellular miRNAs released in PL or serum samples is probably preferable with respect to cell-associated miRNAs as a minimally invasive approach to derive novel biomarkers for clinical use, given the easy accessibility of this fluid. However, because joints are the main targets of clinical manifestations of JIA and the joint environment plays a critical role in disease pathogenesis and progression, miRNA characterization in the SF of affected joints may provide a powerful mean to yield novel early candidate biomarkers of disease development and potential therapeutic targets, and further investigation in this direction is, thus, highly needed. In particular, studies on EV-associated miRNAs need to be boosted up given to the higher stability and detectability of these molecules with respect to ”free-circulating” molecules (see paragraph 1.2) and their better reflection of the disease state, which represent important advantages for new biomarker discovery.

In conclusion, more investigations are demanded to elucidate the potential of miRNAs as biomarkers in JIA. In order to obtain more robust results, authors should take into high consideration the clinical characteristics of patients, the number of patients enrolled, and the techniques for miRNA isolation and analysis.

## 8. Future Directions

Interestingly, studies in Rheumatoid Arthritis (RA) and other chronic inflammatory diseases have outlined the potential of miRNAs as novel markers of the therapeutic efficacy of anti-inflammatory drugs and, in particular, of the prediction of patient response to MTX or inflammatory cytokine inhibitors.

In 2019, Singh and collaborators [123] published the results of a study conducted on 94 DMARD-naïve RA patients in which the expression levels of three selected miRNAs (miR-132, miR-146a, and miR-155) known to play an important role in RA pathology were measured in the whole blood of patients before treatment with MTX. The authors reported significantly lower baseline levels of the three miRNAs in drug-responders compared to non-responders patients. ROC analysis suggested the potential value of all three miRNAs for the prediction of RA patient response to MTX. A placebo-controlled, multicentric, randomized, and double blind phase II study by Daien et al. [124] investigated the safety and efficacy of obefazimod, a first-in-class drug for the treatment of RA that causes the increase of miR-124 expression, known to counteract the inflammatory response through the suppression of several pro-inflammatory cytokine production. Sixty RA patients with active disease resistant to conventional therapy were monitored before and after administration of obefazimod. Significantly higher levels of miR-124 were observed in treated patients when compared to the placebo group, which were associated with a significant decrease in some disease inflammatory indicators, including ESR and CRP, and disease activity, with negligible side effects. According to the authors, these data suggest obefazimod’s efficacy and safety in drug-resistant RA patients. Castro-Villegas investigated by PCR array miRNA levels in the serum of 95 RA patients treated with the anti-TNFa agents, infliximab, etanercept, or adalimumab (55, 25, and 15 patients, respectively) associated with DMARDs for 6 months [125]. Blood samples were collected before and after the 6 months of treatment. miRNA analysis and ROC curves reported by the authors suggest that miR-23 and miR-223 in combination may serve as good predictors of patients’ response to the anti-TNFa/DMARDs therapy determined on the basis of the disease activity score. Krintel and colleagues carried out a double-blind placebo-controlled study on 180 patients with early RA, 89 of whom were treated with the anti-TNFα agent adalimumab and MTX and 91 with placebo- in combination with MTX. Response to treatment was assessed after 12 months by evaluating the disease activity score. The authors reported that patients with low miR-22 and high miR-886-3p expression levels assessed by PCR in the blood before treatment were more prone to achieve a good response to the administration of adalimumab combined with MTX. Thus, the expression levels of these two miRNAs in the blood of RA patients might be predictive of patients’ response to therapy. Ciechomska et al. assessed serum levels of miR-5196 in 10 RA, 13 Ankylosing Spondylitis (AS), and 12 healthy individuals to evaluate its potential as a predictor of anti-TNFα therapy. All patients received MTX in combination with the anti-TNF agents Etanercept (70% of patients) or Adalimumab (30%), and miR-5196 expression was measured by RT-PCR before and after treatment and associated with the response to therapy. The authors observed that the changes in miR-5196 serum expression levels after anti-TNFα therapy were positively correlated with decreased disease activity in both RA and AS patients, suggesting that levels of circulating miR-5196 might be useful for the prediction of anti-TNF therapeutic efficacy. The potential of miR-125a as a predictor of therapy response was investigated in 126 patients with moderate/severe plaque psoriasis treated with etanercept [126]. miR-125a levels measured in the PL of patients at the baseline were lower in responders than in non-responder patients. ROC analysis suggested that PL levels of miR-125a were able to predict patient response to 6-month therapy with etanercept.

Taken together, these studies highlight the high promises of miRNAs as indicators of patient response to specific therapeutics in chronic autoimmune/inflammatory conditions. Since the therapeutic strategies described in this chapter are also applied in pediatric rheumatology, these results may be inspirational for future investigations on miRNAs as therapeutic biomarkers in JIA, which may also help to improve targeted intervention in this disease.

## Figures and Tables

**Table 1 biology-12-00991-t001:** Clinical characteristics, treatment, and biomarkers of JIA subtypes.

Subtype ^a^	Characteristics	Onset Age and Sex	Treatment	Biomarkers ^b^
**Systemic arthritis**	≤10% JIA patients. Number of affected joints ≥ 5. Systemic inflammatory features (continuous fever for at least 2 weeks, rashes, muscle pain, generalized symmetrical lymphadenopathy, enlargement of liver or spleen, or serositis) [12].	Throughout childhood, F = M [1]	Thalidomide (rarely), systemic glucocorticoids, NSAIDs csDMARDs: MTX (with predominant joint inflammation and without active systemic symptoms), bDMARDs: anti-IL-1 (Anakinra, Canakinumab, Rilonacept) [15], anti-IL-6 (Tocilizumab) [15].	S100 proteins IL-18, IL-6, Ferritin [16] RF HLA-B27 ANA Anti-CCP antibodies ESR CRP JADAS [16,17,18]
**Oligoarthritis**	The most common in Western Countries (50–80% cases in Europe and North America) [19]. Number of affected joints within the first six months of disease ≤ 4. High frequency of positivity to ANA. Association with HLA-DRB1*0801 [2].	2–4 y, early childhood, F > > > M [1]	Intrarticular corticosteroid injection(s), NSAIDs csDMARDs: MTX [20], bDMARDSs: TNF inhibitor (Etanercept) [15].	RF HLA-B27 ANA Anti-CCP antibodies ESR CRP JADAS [16,17,18]
Persistent oligoarthritis	Number of affected joints ≤ 4 after the first 6 months from onset [14].			
Extended oligoarthritis	Number of affected joints extends to >4 after the first 6 months of disease [14].			
**Polyarthritis**	Number of affected joints ≥ 5 after the first 6 months from onset, in the absence of fever and in the presence or absence of IgM RF [14].		csDMARDs: MTX, Leflunomide (patients who cannot tolerate MTX), bDMARD: anti-TNF (Etanercept, adalimumab, golimumab), anti-IL-6 (Tocilizumab), anti-CTLA4 (Abatacept-), and Janus kinase (JAK) inhibitors (Tofacitinib) [3].	RF HLA-B27 ANA Anti-CCP antibodies ESR CRP JADAS [16,17,18]
Rheumatoid-factor-positive polyarthritis	5% of JIA cases. Number of affected joints ≥ 5 during the first 6 months of disease. Psitivity for RF [14].	Late childhood or adolescence, F > > M [1]		
Rheumatoid factor negative polyarthritis	15–20% of JIA cases. Number of affected joints ≥ 5 during the first 6 months of disease. Negativity for RF [14].	Early peak 2–4 y and later peak at 6–12 y, F > > M [1]		
**Enthesitis-related arthritis**	Arthritis and enthesitis. Tow or more of the following: presence/history of sacroiliac joint tenderness (with or without lumbosacral pain), HLA-B27 antigen positivity, onset of arthritis in a male >6 years old, acute symptomatic anterior uveitis [14].	Late childhood or adolescence, M > >F [1]	csDMARDs: Sulfasalazine (patients with moderate activity), bDMARDs: TNF inhibitors (Adalimumab, Etanercept) [15].	RF HLA-B27 ANA Anti-CCP antibodies ESR CRP JADAS [16,17,18]
**Psoriatic arthritis**	Characterized by arthritis and psoriasis or arthritis and at least two of the following: dactylitis, nail pitting or onycholysis, psoriasis in a first-degree relative. Psoriasis can occur before or after arthritis onset [21].	Early peak at 2–4 y and later peak at 9–11 y, F > M [1]	csDMARDs: MTX [15], bDMARDs: TNF inhibitors (Adalimumab, Etanercept, Infliximab).	RF HLA-B27 ANA Anti-CCP antibodies ESR CRP JADAS [16,17,18]
**Undifferentiated arthritis**	Not fulfilling any of the listed categories or having criteria formore than one of them.			

^a^ JIA subtypes according to the ILAR classification. ^b^ Main diagnostic and disease activity biomarkers in JIA subtypes. ANA, anti-nuclear antibodies; bDMARDs, biological disease modifying anti-rheumatic drugs; CCP, cyclic citrullinated peptide; csDMARDs, classical disease modifying anti-rheumatic drugs; ESR, erithrocyte sedimentation rate; JADAS, juvenile arthritis disease activity score; NSAIDs, non-steroidal anti-inflammatory drugs; RF, rheumatoid factor.

**Table 2 biology-12-00991-t002:** Summary of the main characteristics of the studies reviewed in the manuscript.

Authors	N° of Patients	Disease Type (N° of Patients)	Types of Samples	Techniques for miRNA Isolation	Techniques for miRNA Analysis	Types of miRNAs
Lashine et al. [100]	10	NA	Peripheral blood mononuclear cells	mirVana miRNA extraction kit	RT-qPCR (TaqMan)	Free circulating
Kamiya et al. [101]	6 24	sJIA (3), pJIA (3) sJIA (8), pJIA (16)	Peripheral blood leukocytes Serum	mirVana PARIS Kit	RT-qPCR (TaqMan)	Free circulating
Schulert et al. [102]	27	sJIA patients: NOS (7) active established (9) CID (11)	Peripheral blood monocytes	mirVana miRNA isolation kit	Microarray	Free circulating
Li et al. [103]	32	sJIA patients:	Peripheral blood monocytes	TRIzol^®^ LS reagent	qPCR (TaqMan)	Free circulating
		NOS (8)				
		established disease (11)				
		CID (13)				
Fan et al. [104]	16	RF-positive pJIA	PBMCs/CD4+ T lymphocytes		qPCR (SYBR Green)	Free circulating
Rajendiran et al. [105]	9	oJIA	Synovial Fluid Mononuclear cells	RNeasy Mini Kit	Gene Chip Array	Free circulating
McAlpine et al. [106]	40	oJIA	Peripheral Blood Leukocytes	miRNeasy Mini Kit	RT-PCR (SYBR Green)	Free circulating
	20		Plasma	miRNeasy Serum/Plasma Kit, ExoRNeasy Serum/Plasma Midi Kit	Digital droplet PCR	Free circulating -Encapsulated
	20		Synovial Fluid	miRNeasy Serum/PlasmaKit, ExoRNeasy Serum/Plasma Midi Kit	digital droplet PCR	Free circulating -Encapsulated
Ma et al. [107]	109	oJIA (43)	Plasma	TRIzol^®^ LS reagent	Microarray-RT-qPCR (SYBR^®^ Green)	Free circulating
		pJIA (37)				
		JAS (29)				
Demir et al. [108]	31	oJIA (17)	Plasma	miRNeasy serum/plasma kit	RT-qPCR (SYBR^®^ Green)	Free circulating
		pJIA (9)				
		ERA (5)				
Sun et al. [109]	155	sJIA (20)	Plasma	TRIzol^®^ LS reagent	Microarray	Free circulating
		JAS (30)				
		SLE (25)				
		KD (40)				
		HSP (40)				
Nziza et al. [110]	34	oJIA (18)	Serum	miRNeasy Serum/Plasma kit	RT-PCR (TaqMan)	Free circulating
		SA (16)	Synovial Fluid			
Raggi et al. [111]	13	oJIA	Plasma	exoRNeasy Serum/Plasma Midi kit	Array Card	Encapsulated
			Synovial Fluid			

## Data Availability

No new data were created or analyzed in this study.

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
