# Peer review of "MicroRNAs in Juvenile Idiopathic Arthritis: State of the Art and Future Perspectives"

_biology, 2023, doi:10.3390/biology12070991_

Round 1
Reviewer 1 Report
Pelassa et al. provide a detailed overview of the current state of literature dealing with the involvement of microRNAs in the pathophysiology of JIA.
Overall, I think the review would improve by shortening and thus making the sections more compact. In some cases, sentences are too long (eg. lines 127-133), and therefore hard to read and understand. Mainly in section 3, comparisons between different studies are distributed throughout the entire section - this should move to the discussion section. Furthermore, the review would profit from providing a graphical abstract.
Minor issues:
- Table 1: Typo in Systemic arthritis not "systenic"
- Table 1: Number of "interested" joints, I suggest the authors wanted to use a different word
Pelassa et al. provide a detailed overview of the current state of literature dealing with the involvement of microRNAs in the pathophysiology of JIA.
Overall, I think the review would improve by shortening and thus making the sections more compact. In some cases, sentences are too long (eg. lines 127-133), and therefore hard to read and understand. Mainly in section 3, comparisons between different studies are distributed throughout the entire section - this should move to the discussion section. Furthermore, the review would profit from providing a graphical abstract.
Minor issues:
- Table 1: Typo in Systemic arthritis not "systenic"
- Table 1: Number of "interested" joints, I suggest the authors wanted to use a different word
Author Response
Pelassa et al. provide a detailed overview of the current state of literature dealing with the involvement of microRNAs in the pathophysiology of JIA.
We thank the referee for appreciating our work.
Overall, I think the review would improve by shortening and thus making the sections more compact. In some cases, sentences are too long (eg. lines 127-133), and therefore hard to read and understand. Mainly in section 3, comparisons between different studies are distributed throughout the entire section - this should move to the discussion section. Furthermore, the review would profit from providing a graphical abstract.
To comply with the request of the reviewer, sections 3, 4, and 5 were made more compact by splitting them into two sub-sections specifically describing miRNA differential expression (3.1, 4.1, and 5.1) and discussing data comparison among different studies (3.2, 4.2, and 5.2) .
For reason of clarity, too long sentences (e.g. that indicated by the reviewer in lines 127-133 of the original version) were shortened or split in two (page.4, lines.131-136; page 12 lines 260-264; page 21 lines 646-652), as suggested.
A higher-quality graphical Abstract was provided.
Minor issues:
- Table 1: Typo in Systemic arthritis not "systenic"
The typo “systenic” in Table 1 was corrected to “systemic”
- Table 1: Number of "interested" joints, I suggest the authors wanted to use a different word
As suggested by the Reviewer, “interested” was changed to “affected”.
Reviewer 2 Report
I have read the review of Pelassa with great interest. It is a detailed overview of the miRNA’s found differently expressed in JIA patients compared to controls. The authors state that this review provides…..and understanding of their role in disease pathogenesis (line 173) . I believe that this is quite overstated in the present form. I do agree that this would be very nice to provide, so allow me to make some suggestions:
-Much of the discussed data is acquired from PBMC/whole blood. I agree that this could result in important biomarker leads, but all differences in miRNA expression could merely be a reflection of cell compositions. Eg, sJIA pt have more neutrophils so you will find more neutrophil-related miRNA’s this does not mean that these miRNA are important in pathogenesis. Please discuss this point about cell composition.
-I would suggest to also discuss the point about activation vs disease. More specifically, microRNA’s in the synovium can be a result of activated immune cells, are they disease specific or contributing? I understand that there might not be specific answers presently, but please discuss the options.
- My main critique are that the targets and functional consequences of the miRNA’s are hardly discussed (this would also be nice to include in a figure). Without that information it’s hard to say anything about the claimed disease pathogenesis. The same holds true for their sentence in line 539 about the novel target for therapy. You have to know what the miRNA’s do and that is hardly discussed. Alternatively you can leave the review as it is but just focus on the associations with expression, and therefore only the function as biomarker.
Author Response
I have read the review of Pelassa with great interest. It is a detailed overview of the miRNA’s found differently expressed in JIA patients compared to controls. The authors state that this review provides…..and understanding of their role in disease pathogenesis (line 173) . I believe that this is quite overstated in the present form. I do agree that this would be very nice to provide, so allow me to make some suggestions:
-Much of the discussed data is acquired from PBMC/whole blood. I agree that this could result in important biomarker leads, but all differences in miRNA expression could merely be a reflection of cell compositions. Eg, sJIA pt have more neutrophils so you will find more neutrophil-related miRNA’s this does not mean that these miRNA are important in pathogenesis. Please discuss this point about cell composition.
-I would suggest to also discuss the point about activation vs disease. More specifically, microRNA’s in the synovium can be a result of activated immune cells, are they disease specific or contributing? I understand that there might not be specific answers presently, but please discuss the options.
- My main critique are that the targets and functional consequences of the miRNA’s are hardly discussed (this would also be nice to include in a figure). Without that information it’s hard to say anything about the claimed disease pathogenesis. The same holds true for their sentence in line 539 about the novel target for therapy. You have to know what the miRNA’s do and that is hardly discussed. Alternatively you can leave the review as it is but just focus on the associations with expression, and therefore only the function as biomarker.
We agree with the Reviewer that the review in the original form poorly addressed the issue of miRNA functions and targets, making hard to draw conclusions about their role in JIA pathogenesis. Because miRNA role in JIA pathogenesis has been reviewed in part by others, as now stated in page 8, lines 194-196 [40;101] and a detailed description of the biologic functions of each of the miRNAs identified in the different studies would increase too much the length of the review (also considering the comment of Reviewer 1 who states that the Review would be improved by shortening), we have decided to remove all the comments on disease pathogenesis and maintain the focus exclusively on miRNA differential expression and role as potential biomarkers in JIA.
Reviewer 3 Report
1. Abbreviation management needs further improvement.
2. The concluding remarks are expected to be original inferences based on the information obtained from the reviewed literature. Therefore, citing articles in the conclusion is not best practice.
Author Response
- Abbreviation management needs further improvement.
To comply with the request of the reviewer, the “Abbreviations” list was included in the text (page 2 lines 43-54) - The concluding remarks are expected to be original inferences based on the information obtained from the reviewed literature. Therefore, citing articles in the conclusion is not best practice.
We agree with the reviewer on this point. References in the Concluding Remarks section were removed
Round 2
Reviewer 2 Report
I believe that the review is improved and I ccan accept it for publication.